# Persistent Hypogammaglobulinemia after Receiving Rituximab Post-HSCT Is Not Caused by an Intrinsic B Cell Defect

**DOI:** 10.3390/ijms242116012

**Published:** 2023-11-06

**Authors:** Lisa M. Ott de Bruin, Ingrid Pico-Knijnenburg, Monique M. van Ostaijen-ten Dam, Thomas J. Weitering, Dagmar Berghuis, Robbert G. M. Bredius, Arjan C. Lankester, Mirjam van der Burg

**Affiliations:** 1Laboratory for Pediatric Immunology, Willem-Alexander Children’s Hospital, Leiden University Medical Center, 2333 ZA Leiden, The Netherlands; l.m.ott_de_bruin@lumc.nl (L.M.O.d.B.); i.pico-knijnenburg@lumc.nl (I.P.-K.); m.m.van_ostaijen-ten_dam@lumc.nl (M.M.v.O.-t.D.); t.j.weitering@lumc.nl (T.J.W.); 2Pediatric Stem Cell Transplantation Program, Willem-Alexander Children’s Hospital, Leiden University Medical Center, 2333 ZA Leiden, The Netherlands; d.berghuis@lumc.nl (D.B.); r.g.m.bredius@lumc.nl (R.G.M.B.); a.lankester@lumc.nl (A.C.L.)

**Keywords:** hematopoietic stem cell transplantation, HSCT, rituximab, RTX, class switch recombination, memory B cells

## Abstract

In the setting of hematopoietic stem cell transplantation (HSCT), Rituximab (RTX) is used for the treatment and prevention of EBV-associated post-transplantation lymphoproliferative disease or autoimmune phenomena such as autoimmune hemolytic anemia (AIHA). Persistent hypogammaglobulinemia and immunoglobulin substitution dependence has been observed in several patients after RTX treatment despite the normalization of total B cell numbers. We aimed to study whether this is a B cell intrinsic phenomenon. We analyzed four patients with different primary diseases who were treated with myeloablative conditioning and matched unrelated donor HSCT who developed persistent hypogammaglobulinemia after receiving RTX treatment. They all received RTX early after HSCT to treat EBV infection or AIHA post-HSCT. All patients showed normalized total B cell numbers but absent to very low IgG positive memory B cells, and three lacked IgA positive memory B cells. All of the patients had full donor chimerism, and none had encountered graft-versus-host disease. Sorted peripheral blood naïve B cells from these patients, when stimulated with CD40L, IL21, IL10 and anti-IgM, demonstrated intact B cell differentiation including the formation of class-switched memory B cells and IgA and IgG production. Peripheral blood T cell numbers including CD4 follicular T-helper (Tfh) cells were all within the normal reference range. In conclusion, in these four HSCT patients, the persistent hypogammaglobulinemia observed after RTX cannot be attributed to an acquired intrinsic B cell problem nor to a reduction in Tfh cell numbers.

## 1. Introduction

Rituximab (RTX) is a chimeric monoclonal antibody targeting CD20-expressing B cells, including pre-B cells in the bone marrow up to mature circulating B cells. For many years, it has been used as treatment for several autoimmune diseases and B cell malignancies. It reduces the number of B cells via the direct cytolytic effects of the Fc receptor, antibody-dependent cytotoxicity and phagocytosis, complement-mediated cell lysis, growth arrest, and B cell apoptosis [1]. Its exact mechanism in treating autoimmune diseases is unclear, as it depletes peripheral CD20-expressing B cells but spares long-lived antibody-producing plasma cells not expressing CD20. This has led to the notion that not only plasma cells or autoantibodies but also other B cell subsets (and their cytokine production) play a role in the pathogenesis of these autoimmune diseases. Indeed, for autoimmune diseases such as multiple sclerosis (mS) and rheumatoid arthritis (RA), clinical improvement after RTX has been shown without a decrease in autoantibody levels [2].

Recently, there has been a growing interest in the phenomenon of persistent low immunoglobulin levels occurring in an undefined subgroup of patients after the use of RTX. Several observational studies showed persistent (>1 year) low immunoglobulin levels (IgG < 5 g/L) after the use of RTX, even though CD20 is not expressed on hematopoietic stem cells, early B cell progenitors (pro-B cells), and long-lived plasma cells. This phenomenon has been reported more frequently in children. In depth analyses of immunological subsets or underlying mechanisms are missing [3,4,5]. Moreover, most studies excluded patients receiving hematopoietic stem cell transplantation (HSCT). In the context of HSCT, RTX is regularly used for the treatment and prevention of EBV-associated post transplantation lymphoproliferative disease (PTLD) and autoimmune hemolytic anemia (AIHA). Several cases of subsequent persistent hypogammaglobulinemia have been reported [6,7,8,9,10].

The production of IgG and IgA depends on effective isotype switching from IgM to IgA and IgG through class switch recombination (CSR). Transitional B cells with a functional B cell receptor (BCR) of the IgM isotype bound to their cell membrane exit the bone marrow and mature into naïve B cells. These naïve B cells circulate the blood and pass through follicles in lymph nodes until their unique receptor recognizes a specific antigen, either soluble or bound to follicular dendritic cells or macrophages. Upon antigen binding to IgM and with additional help from CD4+ follicular T cells, B cells will either differentiate into short-lived extrafollicular plasma cells or memory B cells (mainly IgM positive) or initiate the germinal center (GC) response within the follicle. The GC response is essential for long-lasting high affinity IgA and IgG positive memory B cells and long-lived plasma cells producing high affinity IgA and IgG. The GC B cells undergo clonal expansion while activation-induced (cytidine) deaminase (AID) is expressed to initiate somatic hypermutation of the Ig locus. Upon positive selection of B cells with high affinity antigen binding, CSR is initiated by AID by controlled formation and correct repair of DNA double-strand breaks at conserved motifs within the switch regions, upstream of gene segments that encode distinct constant regions of antibody heavy chains. This results in a class switch from IgM to IgA and/or IgG [11].

Several signals are crucial for the GC response and subsequent CSR: CD40L–CD40 interaction between T and B cells, IL-21 produced by CXCR5+ follicular T helper cells (Tfh), and IL-10 produced by follicular T regulatory cells (Tfr). Therefore, by stimulating naïve B cells with CD40L, anti-IgM, IL-21, and IL-10, it is possible to induce class switch recombination in vitro [12].

In order to better understand the underlying mechanism of the persistent hypogammaglobulinemia that occurred in a subgroup of post-HSCT patients following RTX treatment, we studied the in vitro B cell class switch capacity in four patients and characterized the T cell compartment, including CXCR5+ Tfh cells.

## 2. Results

### 2.1. Patient Characteristics

We identified four patients who had received HSCT for different primary diseases and remained Ig substitution dependent > 5 years after receiving their last dose of RTX, despite reconstituting with normal total B cell counts. All four patients had received the same RTX dose of 375 mg/m^2^. Three patients, UPN 521, UPN 892, and UPN 809, had received two doses with an approximate interval of 1 week. UPN 719 had received one dose, after which the EBV load normalized. UPN 892 received the last dose of RTX 185 days post-HSCT, whereas the others were all given RTX within 30–58 days post-HSCT. Underlying disease, donor type, patient age at the time of HSCT and at the time of the sample date are listed in Table 1. The patients received unmanipulated bone marrow (BM: *n* = 2) and peripheral blood stem cells (PBSC: *n* = 2) from 10/10 matched unrelated donors. The underlying diseases were hemophagocytic lymphohistocytosis (HLH), (pre-B) acute lymphoblastic leukemia (preB-ALL), and hypomorphic RAG1 deficiency. All of the patients showed persistent full donor chimerism in myeloid and lymphoid cells until the last follow-up and did not encounter GvHD or other severe complications. All of the patients showed a normal recovery of total B, T (CD4+ and CD8+), and NK cell counts. The B cell compartment of each patient showed normal naïve B cell counts yet an increased proportion of CD27+/IgM positive (unswitched) memory B cells and severely decreased to absent CD27+/IgG positive (switched) memory B cells. Except for UPN 892, none of the patients had IgA positive-switched memory B cells.

Three patients are currently still Ig substitution dependent (11, 13, and 19 years after their last RTX dose) despite regular attempts to stop Ig suppletion. One patient (UPN 892) showed persistent hypogammaglobulinemia for a period of 6 years after the last RTX before a spontaneous recovery of IgG production occurred. During the 6-year period of Ig substitution, the majority of B cells remained unswitched IgM positive. Thereafter, the percentages of unswitched IgM positive and switched IgG positive memory B cells gradually reached normal values. IgA positive memory B cells were also present but still below normal values. When Ig substitution was discontinued 6 years after the last RTX dose, a normalization of IgG and IgA serum levels occurred.

### 2.2. In Vitro Naive B Cell Stimulation to Induce Class Switch Recombination and Immunoglobulin Production

To study whether the hypogammaglobulinemia and impaired isotype switch observed in vivo was caused by an intrinsic B cell defect, we analyzed this process in vitro using purified naïve B cells stimulated with IL10, IL21, a CD40 ligand, and anti-IgM. Naive B cells from healthy controls and all four patients (UPN 521, 719, 809, and 0892 at the time of hypogammaglobulinemia, Table 1) proliferated and expressed AID at similar levels upon stimulation (Figure 1b,c).

Next, we showed that the sorted naïve B cells of both healthy controls and patients could be equally induced to differentiate into switched IgA and IgG positive memory B cells after 6 days of in vitro stimulation (Figure 2). After 10 days of in vitro stimulation, IgA and IgG were detected at similar levels in the supernatant of both patient and healthy control B cell cultures, while they were absent in the unstimulated wells (Figure 3). Together, these results suggest an intact function and differentiation capacity of naïve B cells in the four patients at the time when they suffered from severe hypogammaglobulinemia (Table 1).

### 2.3. CXCR5+ Tfh Cells

Because these patients are unable to produce sufficient IgG (and IgA) in vivo, while their cells are able to undergo CSR and produce IgG and IgA in vitro when stimulated with CD40L, IL10, IL21, and anti-IgM, we studied the presence of circulating Tfh cells (CD3^+^CD4^+^CXCR5^+^), which are essential for class switch and IgG production in vivo. In all four patients, Tfh cells were present within the normal age-corrected range (Table 1). In UPN 892, the Tfh cell numbers were within the normal age-corrected range during both the hypogammaglobulinemia phase as well as after spontaneous recovery from hypogammaglobulinemia [13].

## 3. Discussion

Persistent hypogammaglobulinemia after rituximab (RTX) treatment is a known phenomenon, especially in children. The underlying mechanism is unresolved, and only observational studies have been published so far that mostly concern non-HSCT patients [3,4,5,14,15,16]. In the non-HSCT population, a persistent IgM deficiency was reported most frequently after RTX treatment [1,3]. This could possibly be explained by the fact that long-lived plasma cells without surface CD20 mainly produce IgA and IgG, whereas short-lived CD20-expressing plasma blasts mainly produce IgM. An IgA deficiency is least frequently reported, possibly due to RTX resistant mucosal IgA^+^ plasma cells and IgA^+^ long-lived plasma cells in the bone marrow. However, in these observational studies, a detailed analysis of the B cell compartment is missing.

In recent years, several cases have been published, demonstrating persistent hypogammaglobulinemia after RTX treatment in HSCT patients—in some cases, lasting even more than 10 years after HSCT and RTX treatment, despite normal B cell counts and IgM levels [6,7,8,9,10,17]. In our pediatric HSCT cohort, we identified four patients who had received RTX post-HSCT for either immune cytopenia or EBV reactivation and who developed persistent hypogammaglobulinemia and immunoglobulin supplementation dependency. In all four cases, very low numbers of switched memory B cells were observed, while unswitched/natural effector/marginal zone B cells (CD19^+^CD27^+^IgD^+^) were reduced in three of the four patients, which is similar to a recent report by Marzollo et al. [18].

Previous in vitro studies showed preferential depletion by RTX of unswitched B cells, possibly explaining the decreased unswitched population [19]. All four of our selected patients similarly showed mainly IgM^+^-switched memory B cells and no or severely reduced IgG^+^ memory B cells. Yet, when the sorted naïve B cells of these patients were stimulated in vitro with IL10, IL21, a CD40 ligand, and anti IgM, we observed proliferation, AID expression, and differentiation into IgG^+^ and IgA^+^ memory B cells and eventually IgA and IgG production at similar levels as compared to healthy controls, indicating a preserved intact intrinsic B cell function.

Differentiation into switched memory B cells (CD19^+^CD27^+^IgD^−^) in vivo requires GC formation and T helper cell signals to induce CSR and somatic hypermutation. Therefore, we studied the presence of (CD3^+^CD4^+^CXCR5^+^) circulating Tfh cells, which are considered to mirror their presence in the GC [20]. In all four patients, the absolute number of Tfh cells was within the normal age-adjusted range. A limitation of this analysis is that the peripheral blood Tfh population may not fully reflect its presence and functional integrity in the GC; therefore, an analysis of lymph node tissue and other lymphoid organs that assesses the T–B cell interaction in more detail would certainly be of added value. Unfortunately, and for obvious ethical reasons, lymphoid tissue could not be obtained for research purposes in these children. To date, it remains unclear how after six years of hypogammaglobulinemia, UPN 892 spontaneously recovered with switched memory B cells and concomitant normalization of immunoglobulin IgG production and adequate antibody titers after vaccination with both conjugate and polysaccharide vaccines allowing discontinuation of immunoglobulin supplementation

## 4. Materials and Methods

### 4.1. Patient Selection

At the LUMC-WAKZ, with the approval of the Institutional Review Board (protocols P00.068, P01.028, and B17.001), peripheral blood mononuclear cell (PBMC) samples of each patient (recipient) and healthy donors undergoing HSCT were longitudinally stored at the LUMC biobank after informed consent was obtained. For this study, four patients were included who received an HSCT and remained on immunoglobulin (Ig) substitution more than a year after receiving their last dose of RTX, despite normal B cell counts, full donor chimerism and no signs of graft versus host disease (GvHD).

Four healthy controls were obtained from the LUMC healthy voluntary donor service (LUVDS). The LuVDS Biobank was stored in the central biobanking facility at the LUMC (LuVDS-DG L21.002), and direct use was coordinated by the same entity. 

### 4.2. In Vitro Stimulation of Naïve B Cells

The different steps that were taken are summarized in Figure 1a. Naïve B cells (CD20^+^CD27^−^IgG^−^IgA^−^) were sorted on the FACS Aria III (BD Biosciences, San Jose, CA, USA) from the cryopreserved peripheral blood mononuclear cells (PBMCs) of patients and healthy donors. The following fluorochrome-conjugated antibodies were used: CD20 (Pe-cy7, clone L27, BD), CD27 (APC, clone L128, BD), IgG (PE, clone G18-145, BD), and IgA (FITC, clone IS11-8E10, Miltenyi, Bergisch Gladbach, Germany).

The sorted cells were cultured for 6 or 10 days in round bottom 96-well plates (5 × 10^4^/well) in a RPMI1640 medium supplemented with 10% FCS, penicillin/streptomycin, 50 mM b-mercaptoethanol (Sigma-Aldrich, St. Louis, MO, USA), 2.5 mg/mL transferrin (Fluka, Buchs, Switzerland), 1 mg/mL insulin (Sigma-Aldrich), and non-essential amino acid (Gibco, Billings, MT, USA). The cells were stimulated with a 100 ng/mL CD40 ligand (Enzo, Farmingdale, NY, USA), 100 ng/mL IL-21 (Peprotech, Rocky Hill, NJ, USA), 50 ng/mL IL-10 (Peprotech) and 10 mg/mL anti-IgM (Jackson Immunoresearch, West Grove, PA, USA), as previously published [12].

### 4.3. The Expression of AID by Real-Time Quantitative RT-PCR (RT-qPCR)

After 6 days cells were counted, the total RNA was extracted from the cultured cells, and activation-induced (cytidine) deaminase (AID) expression was measured by qPCR, as previously described [21]. Briefly, RNA was isolated from unstimulated and stimulated naive B cells after 6 days of culture using the RNeasy Micro Kit (Qiagen, Hilden, Germany) and subjected to reverse transcription with SuperScript II Reverse transcription (Invitrogen, Waltham, MA, USA). A 2× GoTaq qPCR Master Mix (Promega, Madison, WI, USA) was used together with the previously described primers for the amplification of AID transcripts [13]. AID expression was normalized to the housekeeping gene GUSB.

### 4.4. Class Switching Analyzed by Flow Cytometry and ELISA

On day 6, class switching was analyzed via flow cytometry. DAPI was added to discriminate between live and dead cells. The following fluorochrome-conjugated antibodies were used: CD19 (PC7, clone J3-119, Beckman Coulter, Brea, CA, USA), IgA (FITC, clone is 11-8E10, Miltenyi), IgG (PE, clone G18-145, BD), IgM (BV510, clone MHM-88, Biolegend, San Diego, CA, USA), IgD (PerCP-Cy5.5, clone IA6-2, Biolegend), and CD27 (APC, clone L128, BD, CD38 (APC-AF750, LS198-4-3, Beckman Coulter).

On day 10, the supernatants were collected, and IgG and IgA levels were measured with ELISA using alkaline phosphatase-conjugated goat anti-human IgG or IgA (Invitrogen) for detection. Absorbances were measured on the VersaMax (Molecular Devices, San Jose, CA, USA).

### 4.5. T and B Cell Subset Analysis

PBMCs from the same thawed sample were used for elaborate B and T cell subset analysis; the cells were washed and stained with a panel of fluorochrome-conjugated monoclonal antibodies (Appendix A) and brilliant stain plus buffer (BD Biosciences) and were subsequently diluted in FACS buffer. Samples were measured on a 5-laser spectral flow cytometry (Cytek Aurora, Cytek Biosciences, Fremont, CA, USA) at the Flow Cytometry Core Facility (FCF) of the LUMC. Measurements and spectral unmixing were performed using Spectroflo^®^ Software v2.1.0 (Cytek Biosciences). Data were analyzed and visualized using the cytometry analysis platform OMIQ (Omiq, Boston, MA, USA) (Gating strategy, Appendix A).

## 5. Conclusions

In conclusion, we demonstrate in four HSCT patients with persistent hypogammaglobulinemia after RTX, that this secondary immunodeficiency cannot be attributed to an acquired intrinsic B cell problem nor to a reduction in Tfh cell numbers. Our data point to an impaired non-B cell intrinsic differentiation and isotype switching defect. A further analysis of the B cell response in the lymphoid tissue and GC formation is required to unravel the mechanism for RTX-induced persistent hypogammaglobulinemia.

## Figures and Tables

**Figure 1 ijms-24-16012-f001:**
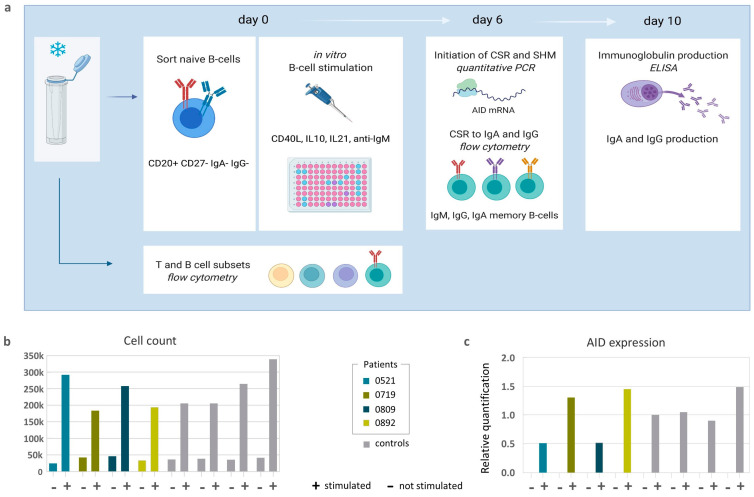
(**a**) A schematic overview of the procedure. Cryopreserved PBMCs were taken from a biobank. Part of the sample was used for the flow cytometry of T and B cell subsets. The rest of the sample was used to FACS sort naïve B cells (CD20^+^CD27^−^IgA^−^IgG^−^) to stimulate in vitro. On day 6, part of the cells were counted, RNA was isolated for AID expression, and IgA and IgG memory B cells were measured via flow cytometry. The rest of the cells were cultured until day 10 when the supernatant was collected for ELISA in order to measure IgA and IgG production; (**b**) Absolute cell counts on day 6 after stimulation compared to unstimulated wells; (**c**) AID expression was normalized to the housekeeping gene GUSB.

**Figure 2 ijms-24-16012-f002:**
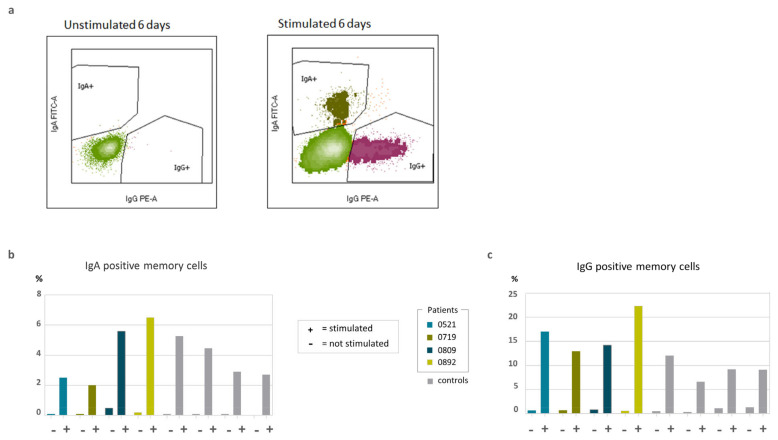
(**a**) Flow cytometry gating strategy on day 6. (**b**) Percentage of IgA positive memory B cells and (**c**) percentage of IgG positive memory B cells. Stimulated (+) versus unstimulated (−) wells.

**Figure 3 ijms-24-16012-f003:**
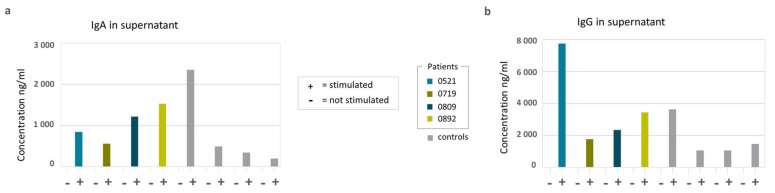
ELISA on collected supernatant on day 10. (**a**) IgA in ng/mL; (**b**) IgG in ng/mL.

**Table 1 ijms-24-16012-t001:** The four selected patients with their characteristics and lymphocyte subsets.

UPN	521	719	809	892	
**Underlying disease**	primary HLH	preB-ALL	RAG1 deficiency	primary HLH	
**Donor type**	10/10 MUD-PBSC	10/10 MUD-BM	10/10 MUD-BM	10/10 MUD-PBSC	
**Age (in years) at SCT**	2.9	2.1	3.1	1.4	
**Number of RTX dosages (375 mg/m^2^)**	2	1	2	2	
**Last RTX dose, days after SCT**	58	28	30	185	
**Indication**	EBV	EBV	AIHA+EBV	AIHA	
**Current duration hypogammaglobulinemia**	19 years	13 years	11 years	recovery after 6 years	
				*IVIG dep.*	*IVIG indep.*
**Age (in years) at sample date**	21.5	2.9	12.4	3.3	7.8
**IgA**	<0.06	<0.06	<0.04	0.52	0.51
**IgM**	1.76	0.68	9.99	0.52	2.08
**IgG**	6.5 *	3.57	9.7 *	2.72	7.8
**B cells** (CD19^+^)	269	438	215	325	450
**B transitional** (CD38^hi^/CD24^hi^)	1 ↓	33	22	21 ↓	3 ↓
**B naive mature** (CD38^dim^/CD24^dim^/IgD^+^/CD27^−^)	213	343	165	269	383
**B natural effector** (CD38^dim^/IgD^+^/CD27^+^)	15	9 ↓	2 ↓	6 ↓	21
**B memory** (CD38^dim^/IgD^−^/CD27^+^)	14	12 ↓	5 ↓	3 ↓	14
**B memory IgM**	86 ↑	92 ↑	93 ↑	88 ↑	67 ↑
**B memory IgG**	11 ↓	0 ↓	0 ↓	0 ↓	23
**B memory IgA**	0 ↓	0 ↓	0 ↓	6	2 ↓
**NK** (CD56^+^)	97	207	146	131	534
**T cells** (CD3^+^)	1962	1340	1835	4161	2703
**CD4^+^ T cells** (T4)	894	600	860	1115	1097
**TfH** (CXCR5^+^)	89	49	30	21	59
**CD8^+^ T cells**	834	641	691	2311 ↑	1227

HLH = Hemophagocytic lymphohistiocytosis; preB-ALL = pre-B cell acute lymphatic leukemia; MUD = matched unrelated donor; BM = bone marrow; PBSC = peripheral blood stem cells; Cell numbers shown in cells/µL, immunoglobulin levels in g/L. IgM, IgG and IgA memory B cells as a percentage of total switched memory B cells. Reference range for age is used. * On IVIG suppletion. ↑ above ref. range for age, ↓ below ref. range for age, cells/µL.

## Data Availability

Data are contained within the article and Appendix A.

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
