# Peer review of "Persistent Hypogammaglobulinemia after Receiving Rituximab Post-HSCT Is Not Caused by an Intrinsic B Cell Defect"

_ijms, 2023, doi:10.3390/ijms242116012_

Round 1
Reviewer 1 Report
Comments and Suggestions for Authors
In this article, the authors aimed to investigate the reasons for persistent hypogammaglobulinemia observed in four patients. By demonstrating the in-vitro differentiation capacity of IgM B-cells from these patients to IgA and IgD expressing cells, they showed that the class switching deficiency observed in patients is not a cell intrinsic deficiency. This finding is indeed interesting. However, I noticed that none of the figures include any error bars, which suggests that the experiments may have been conducted only once. Indeed, replicating the experiments three times would further strengthen the study.
Author Response
We understand it would be better to test each patient sample in three different experiments. Unfortunately, we were limited by patient material and there is no patient material left to repeat the experiments a third time.
Reviewer 2 Report
Comments and Suggestions for Authors
The authors revealed that the persistent hypogammaglobulinemia after receiving RTX in HSCT patients is not caused by an acquired intrinsic B cell defect or a reduction in Tfh cell numbers. Although the sample volume is limited (only four patients), the flow cytometry experiments are designed appropriately and the data is carefully analyzed and presented. More importantly, the results are significant to the application of the RTX, providing insights into future research.
Please add a legend to Figure 3.
Comments on the Quality of English Language
The English quality is fine. No significant issue is detected.
Author Response
We thank the reviewer for pointing out the missing legend in figure 3. It has been added now.